# Emergent normal fluid in the superconducting ground state of overdoped cuprates

Shusen Ye[1], Miao Xu[1], Hongtao Yan[2], Zi-Xiang Li[2,3,4], Changwei Zou[1], Xintong Li[1,2], Zhenqi Hao[1], Chaohui Yin[2], Yiwen Chen[2], Xingjiang Zhou[2], Dung-Hai Lee ●[3,4] & Yayu Wang ●[1,5,6] ✉

The microscopic mechanism for the disappearance of superconductivity in overdoped cuprates is still under heated debate. Here we use scanning tunneling spectroscopy to investigate the evolution of quasiparticle interference phenomenon in $Bi_2Sr_2CuO_{6+\delta}$ over a wide range of hole densities. We find that when the system enters the overdoped regime, a peculiar quasiparticle interference wavevector with arc-like pattern starts to emerge even at zero bias, and its intensity grows with increasing doping level. Its energy dispersion is incompatible with the octet model for $d$-wave superconductivity, but is highly consistent with the scattering interference of gapless normal carriers. The gapless quasiparticles are mainly located near the antinodes and are independent of temperature, consistent with the disorder scattering mechanism. We propose that a branch of normal fluid emerges from the pair-breaking scattering between flat antinodal bands in the quantum ground state, which is the primary cause for the reduction of superfluid density and suppression of superconductivity in overdoped cuprates.

One of the main puzzles concerning the cuprate high-temperature superconductors is the dome-shaped doping dependence of the critical temperature ($T_c$)[1]. The increase of $T_c$ in the underdoped regime is widely believed to be due to the increase of superfluid density hence phase stiffness by doping the parent Mott insulator, as illustrated by the Uemura law[2]. In the overdoped regime, the mechanism for the disappearance of superconductivity at the quantum superconductor to metal transition (QSMT) is still elusive[3]. It was usually considered to be due to the closing of superconducting gap like in conventional superconductors, but there are growing evidence that the gradual losing of phase coherence plays a crucial role in the QSMT[4–7]. Especially, mutual inductance experiment reveals that overdoped cuprates exhibit an anomalous missing of superfluid density[4,5,8], which is intimately related to the decrease of $T_c$. The reduced phase stiffness

results in strong phase fluctuation of order parameter and gradual gap filling across $T_c$, as demonstrated by angle-resolved photoemission (ARPES) experiments[6]. On the theoretical front, it has been proposed that the extensive pair-breaking scattering between the flat antinodal bands neighboring the van Hove singularity (vHS) is responsible for the suppression of phase stiffness[9–11]. When the disorder scattering strength is increased to $\hbar/\tau \geq \Delta_{SC}$, where $\Delta_{SC}$ is the superconducting gap and $\tau$ is the quasiparticle lifetime, pair-breaking scattering is expected to generate finite density of state (DOS) at the Fermi energy ($E_F$)[12–14]. Considering the conservation of electronic spectral weight, the unpaired electrons would emerge from the extensive pair-breaking process and coexist with the superconducting condensate. The nature of such quantum normal fluid has yet to be demonstrated experimentally in detailed.

[1]State Key Laboratory of Low Dimensional Quantum Physics, Department of Physics, Tsinghua University, Beijing, P. R. China. [2]Beijing National Laboratory for Condensed Matter Physics, Institute of Physics, Chinese Academy of Sciences, Beijing, P. R. China. [3]Department of Physics, University of California, Berkeley, CA, USA. [4]Materials Sciences Division, Lawrence Berkeley National Laboratory, Berkeley, CA, USA. [5]New Cornerstone Science Laboratory, Frontier Science Center for Quantum Information, Beijing, P. R. China. [6]Hefei National Laboratory, Hefei, P. R. China. ✉e-mail: yayuwang@tsinghua.edu.cn

Scanning tunneling microscopy (STM) is a powerful technique to probe the normal fluid in the superconducting ground state owing to its ability to directly measure the single electron density of state (DOS) and image the characteristic scattering interference of normal carriers. Previous STM studies on overdoped cuprates have unveiled the restoration of a large hole-type Fermi surface[15–17], the closing of pseudogap[18], and the diminishing of checkerboard charge order[19,20]. Recently, the spatially periodic and particle-hole asymmetric modulation of the coherence peaks have been observed in overdoped $Bi_2Sr_2CaCu_2O_{8+\delta}$ (Bi-2212) and $Bi_2Sr_2Ca_2Cu_3O_{10+\delta}$ (Bi-2223), and is shown to be consistent with the strong pair-breaking antinode-antinode scattering induced by disorders[10]. However, due to the large antinodal $\Delta_{SC}$ in Bi-2212 and Bi-2223, the normal fluid generated by such pair-breaking has not been observed because the condition $\hbar/\tau \geq \Delta_{SC}$ is difficult to meet. In contrast, the single-layer compound $Bi_2Sr_2CuO_{6+\delta}$ (Bi-2201) has much smaller antinodal $\Delta_{SC}$ and widely tunable doping levels[21], providing an ideal system for investigating the emergent normal fluid and QSMT in the overdoped regime[3,9].

In this work, we use STM to investigate the evolution of quasiparticle interference (QPI) phenomenon in Bi-2201 over a wide range of hole densities. When the system enters the overdoped regime, an antinodal arc-like QPI wavevector starts to emerge even at zero bias. Its energy dispersion and temperature dependence are incompatible with the octet model for superconducting Bogoliubov quasiparticles with $d$-wave pairing symmetry, but are highly consistent with the scattering interference of gapless normal carriers. The emergent normal fluid in the overdoped superconducting ground state suggests that the sign-changing pair-breaking scattering between antinodal quasiparticles is the primary cause for the reduction of superfluid density and suppression of $T_c$ in overdoped cuprates[9].

## Result

### Zero-bias density of state

Figure 1a shows the schematic phase diagram of Bi-2201 with the circles indicating the seven samples studied in this work denoted as UD-20K, OP-32K, OD-28K, OD-24K, OD-15K, OD-3K, and OD-NSC, respectively. Here UD/OP/OD represents underdoped/optimally doped/overdoped, the number represents $T_c$, and NSC represents non-superconducting. The hole densities are estimated to be $p = 0.13, 0.16, 0.18, 0.19, 0.21, 0.22$ and $0.23$, respectively, by the empirical $T_c$ versus $p$ relation described in Supplemental Material Section I. Figure 1b displays the spatially averaged $dI/dV$ spectra normalized by the mean DOS values at $\pm70$ meV (see Supplementary Material Section II for more discussions), which demonstrates the evolution from the coexistence of superconducting gap (-9 meV) and pseudogap (-39 meV) in OP-32K[16,19] to a vHS peak in the OD-NSC sample[17]. Despite the microscopic

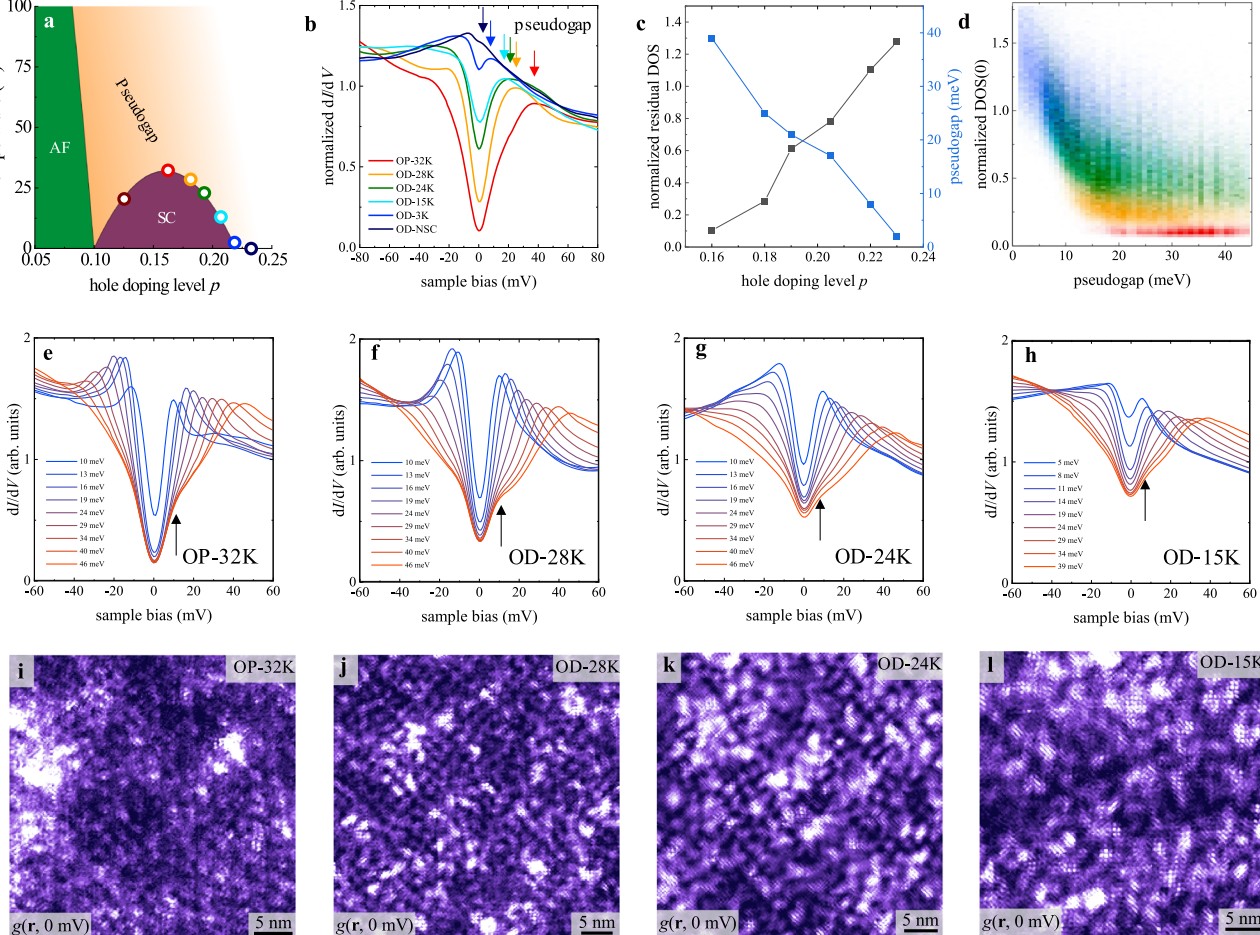

**Fig. 1 | The spatial averaged spectra of all samples. a** The schematic phase diagram of Bi-2201. The open circles indicate the hole densities of the seven samples studied in this work. **b** Spatially averaged $dI/dV$ spectra on the optimally doped and overdoped samples normalized by the mean DOS at $\pm70$ mV. The pseudogap position of each curve is marked by an arrow with the corresponding color. The normalization points are set at $\pm70$ mV because the spectrum exhibits rather smooth features at these high energies, much larger than the pseudogap size. The setup parameters are enumerated in Tab. S2. **c** With increasing hole density in the overdoped regime, the pseudogap size decreases systematically while the normalized zero-bias DOS increases. **d** Heatmap of the pseudogap and normalized zero-bias DOS extracted from the spatially resolved $dI/dV$ spectra over a large area in each overdoped superconducting sample. **e–h** The averaged spectra were sorted by the pseudogap size in the OP-32K, OD-28K, OD-24K, and OD-15K samples. **i–l** The conductance map measured at zero bias for each sample in **e-h**.

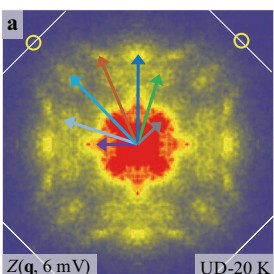
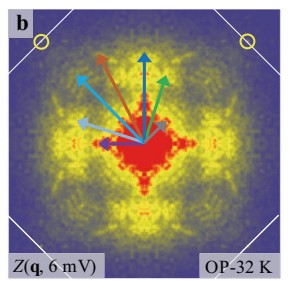
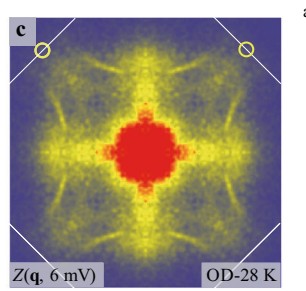
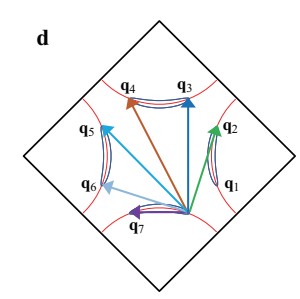

**Fig. 2 | Bogoliubov QPI near optimal doping. a–c** FT of the conductance ratio map $Z(\mathbf{q}, 6\,\text{mV})$ on the UD-20K, OP-32K and OD-28K samples, respectively. The seven arrows indicate the dominant wavevectors, which are consistent with the octet model for Bogoliubov QPI with $d$-wave symmetry. The yellow circles display the Bragg wavevector and the white lines indicate the Brillouin zone boundaries. **d** Schematic diagram of the octet model with seven scattering wavevectors.

inhomogeneities in each sample, the spatially averaged spectra still provide the representative electronic structures of the whole samples that evolve systematically with doping level. In addition to the systematic decrease of pseudogap size $\Delta_{\text{PG}}$, another trend is the nearly linear increase of residual DOS at $E_F$ (or zero bias) with overdoping, as summarized in Fig. 1c. The increase of residual DOS at low temperature is consistent with the increasing residual electronic specific heat in overdoped cuprates[22–25]. To reveal the evolution of zero-bias DOS within each sample, we use clustering analysis to classify the large number of d$I$/d$V$ spectra obtained on a spatial grid by the pseudogap size, as shown in Fig. 1e–h for the OP-32K, OD-28K, OD-24K, and OD-15K samples. Within each sample, the zero-bias DOS always increases with reducing pseudogap size, hence the local hole density. The trend becomes more pronounced in the more overdoped samples, and the zero-bias DOS in the underdoped sample varies little with pseudogap size (see Supplementary Material Section III for details). With the continuous increase of zero-bias DOS, the superconducting gap features are evident in the large pseudogap spectra of each sample, as indicated by the arrows in Fig. 1e–h[26,27]. In Fig. 1i–l, the corresponding zero-bias DOS maps reveal the regular checkerboard order that diminishes into short-range glassy patterns with overdoping[19]. The heatmap in Fig. 1d directly visualizes the distribution of residual DOS versus $\Delta_{\text{PG}}$ in the five overdoped superconducting samples.

## Weakened octet-model QPI in the overdoped regime

To elucidate the nature of low energy excitations, we obtain $g(\mathbf{r}, E) = $ d$I$/d$V(\mathbf{r}, E)$ over a large area in each sample, and use the Fourier transform (FT) map $g(\mathbf{q}, E)$ to identify the QPI wavevectors. To enhance the Bogoliubov QPI patterns due to superconductivity, we use the conductance ratio map $Z(\mathbf{r}, E) = g(\mathbf{r}, E)/g(\mathbf{r}, -E)$ to eliminate the influence of the setpoint effect and enhance the particle-hole symmetric superconducting Bogoliubov QPI[28]. The FT of Bogoliubov QPI patterns on the UD-20K and OP-32K samples are shown in Fig. 2a, b with seven arrows indicating the dominant wavevectors, which are consistent with the octet model for $d$-wave superconductivity as sketched in Fig. 2d[29]. Here the scattering mainly occurs between the hot spots of the banana-shaped equal-energy contours of the Bogoliubov quasiparticles[28,30,31]. With increasing doping levels, the Bogoliubov QPI is gradually weakened, as reported by earlier studies on overdoped cuprates[32]. For the slightly overdoped OD-28K sample in Fig. 2c, the octet QPI wavevectors become nearly invisible. The weakening of octet-model QPI was also observed previously in overdoped Bi-2201[16,19], which may be ascribed to the presence of strong phase fluctuation.

## Zero-bias QPI of the normal fluid in the overdoped regime

Next, we focus on the QPI phenomenon from the normal carriers by exploring the conductance map at zero bias. The Bogoliubov

quasiparticles of a superconductor are gapped out at $E_F$, and thus do not contribute to the zero-bias QPI. Therefore, the FT of zero-bias conductance map $g(\mathbf{q}, 0\,\text{mV})$ directly reveals the QPI originating from the normal carriers at $E_F$. The zero-bias QPI in the OD-NSC sample $g(\mathbf{q}, 0\,\text{mV})$ exhibits pronounced features consisting of arcs highlighted by the yellow line in Fig. 3a. As described in a previous report[33,34], such patterns can be well reproduced by the autocorrelation (Fig. 3b) of the single hole-like Fermi surface in Fig. 3c. For the superconducting ground state with well-defined $d$-wave pairing symmetry and large antinodal gap size, the antinodal Fermi surface is completely gapped out, thus cannot contribute to the zero-bias QPI. As expected, in the OP-32K sample shown in Fig. 3d, the arc-like features are absent in $g(\mathbf{q}, 0\,\text{mV})$.

However, we find unexpectedly that even for the slightly overdoped OD-28K sample, the $g(\mathbf{q}, 0\,\text{mV})$ map shown in Fig. 3e already exhibits the arc-like QPI feature (the transversal replica is due to the structural supermodulation). With increasing doping in Fig. 3e–h, the zero-bias arc-like QPI features become more pronounced, and the tip of the circle moves closer toward the vHS point. The striking resemblance to the OD-NSC sample, especially the presence of QPI at zero bias, strongly suggests that an electronic component with a normal quasiparticle starts to emerge in the ground state when Bi-2201 enters the overdoped regime. In Fig. 3i, we extract the Fermi surface of each sample following the same method for the OD-NSC sample[17], which displays a continuous expansion of the hole pocket with increasing $p$, providing further evidence for the normal carrier origin of the arc-like QPI. We then extract the QPI intensity near the antinode by integrating the $g(\mathbf{q}, 0\,\text{mV})$ along the Fermi wavevector at the antinodal region (see Supplementary Material Section V for details), since the zero-bias QPI is mainly located at the antinodal region. As shown in Fig. 3j, the normal carrier QPI intensity grows with increasing doping and anti-correlates with $T_c$, which provides more direct evidence of increasing normal carrier at antinodal region than $k$-integrated the zero-bias DOS in Fig. 1b. We emphasize that the arc-like normal carrier QPI at zero bias is totally different from the Fermi arc in underdoped cuprates observed by ARPES[35], which is centered around the nodal points and caused by the pseudogap phenomenon. Here the normal carriers are mainly concentrated at the antinodal region and emerge with the overdoping process.

Figure 4a–f shows the QPI patterns measured at $V = -50, -30, -10, 10, 30$ and $50\,\text{mV}$ on the OD-15K sample, including energies inside and outside the pseudogap, respectively. The patterns match well with the equal-energy contours of the normal carrier band structure obtained by the tight-binding model, as shown in Fig. 4g. For $V = -50\,\text{mV}$, the band topology changes to that of a closed electron-type pocket near the antinodal region because now the energy lies below the vHS. To further reveal the bias dependence of QPI, in Fig. 4h, we plot the normalized cross-sectional $g(q_y, E)$ with fixed $q_x = \frac{2}{3}Q_{\text{Bragg}}$, as

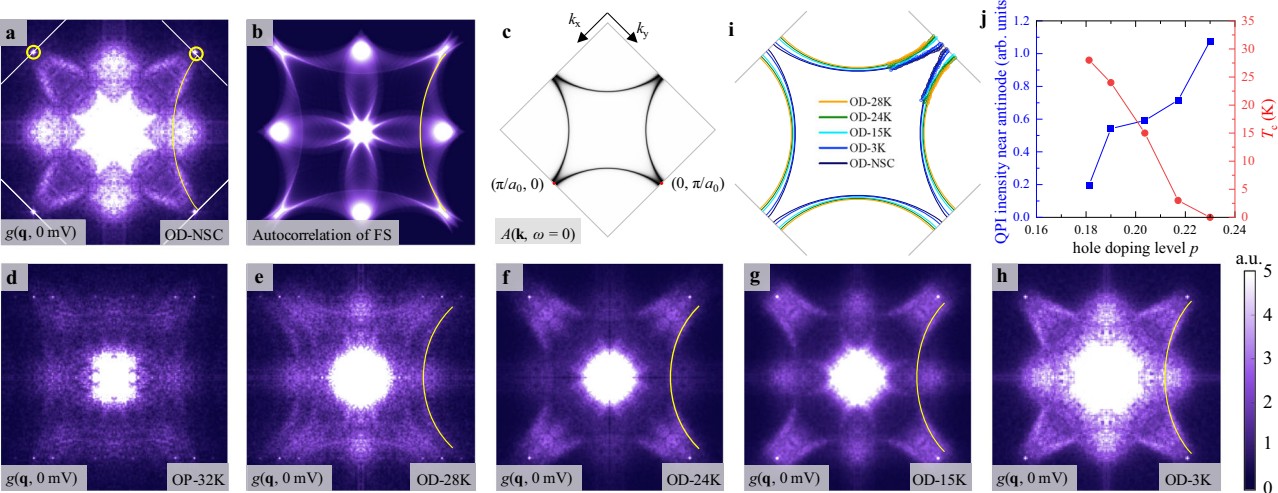

**Fig. 3 | The arc-like QPI at zero-bias in overdoped Bi-2201. a** The zero-bias $g(\mathbf{q}, 0\,\text{mV})$ map of the OD-NSC sample. The arc-like pattern reflects the QPI of the hole-type Fermi surface. **b** Autocorrelation of the Fermi surface calculated by the tight-binding model, which nicely reproduces the pattern in (**a**). **c** The hole-type spectral function at $E_\text{F}$ is used in (**b**). **d**–**h** The $g(\mathbf{q}, 0\,\text{mV})$ maps on the OP-32K, OD-28K, OD-24K, OD-15K, and OD-3K samples, respectively. All the overdoped samples exhibit the arc-like feature highly resembling the normal state Fermi surface in the OD-NSC sample. **i** The Fermi surface extracted from the $g(\mathbf{q}, 0\,\text{mV})$ maps in the overdoped samples, revealing the increase of Luttinger volume with overdoping. **j** The doping dependence of QPI intensity near antinodes in all overdoped samples with the respective $T_\text{c}$. The complete dataset of the topography and real-space DOS maps is displayed in Supplementary Material Section IV.

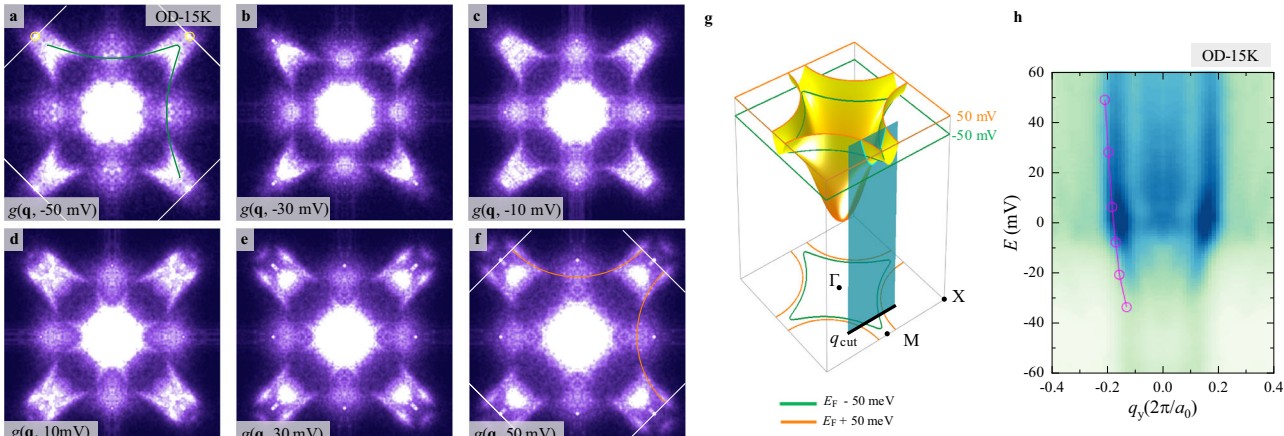

**Fig. 4 | The energy-dependence of normal QPI on the OD-15K sample. a**–**f** FT of the conductance maps $g(\mathbf{q}, E)$ at $E = -50, -30, -10, 10, 30,$ and $50\,\text{mV}$, respectively, on the OD-15K sample. The green and orange lines are equal-energy contours based on the tight-binding model at $-50$ and $50\,\text{mV}$. **g** Schematic diagram of the band dispersion. The guidelines in **a** and **f** are sketched with the same colors and projected to the bottom plane. **h** The intensity of QPI on the cross-section near the antinodal region is indicated by the cyan plane in **g** with open circles indicating the local maximum of QPI intensity. The dispersion demonstrates that the high-energy contours continuously extend to zero bias within the superconducting gap.

illustrated by the cyan plane in Fig. 4g. It shows electron-type parabolic dispersion near the vHS at the M point, and the same plot for other overdoped samples exhibit systematic variations (see Supplementary Material Section VI for details). More interestingly, it is apparent that the strongest QPI signals for all samples are located at $E_\text{F}$, which is strong evidence that they originate from normal carriers on the Fermi surface, as expected from the energy-dependent lifetime of Landau quasiparticles. Similar arc-like QPI pattern is also observed at finite energies of the UD-20K and OP-32K samples (see Supplementary Material Section VII for details), which is consistent with the shift of normal carrier band structure with overdoping. In fact, QPI with a similar pattern has been observed in the conductance ratio map at finite energy[16]. Furthermore, the aforementioned dataset completely reveals the nature of the quasiparticle, containing a branch of dispersive normal carrier beyond the octet-model QPI of Bogoliubov quasiparticles.

## The spatial inhomogeneity of the normal carrier QPI

It may be argued that the normal carrier QPI in the superconducting ground state of overdoped cuprates comes from normal regions without superconductivity due to strong spatial inhomogeneity. To check if this is the case, we use the local pseudogap size related to the spatial inhomogeneity of the doping level, following a well-established practice[20]. The spatial distribution of $\Delta_\text{PG}$ in the OD-15K sample is displayed in Fig. 5a, which exhibits the usual nanoscale patches. We carry out clustering analysis of the whole field of view according to the local pseudogap size and obtain the FT of zero-bias QPI on each part. The FT images within the smallest averaged pseudogap (7 meV) and the largest averaged pseudogap (29 meV) areas are displayed in Fig. 5b, c, and the other areas with moderate pseudogap size are enumerated in Supplementary Material Section VIII. The arc-like QPI exists in the $g(\mathbf{q}, 0\,\text{mV})$ of every regime, but with relatively weaker intensity in larger pseudogap area, as summarized in Fig. 5d, which is the same as the

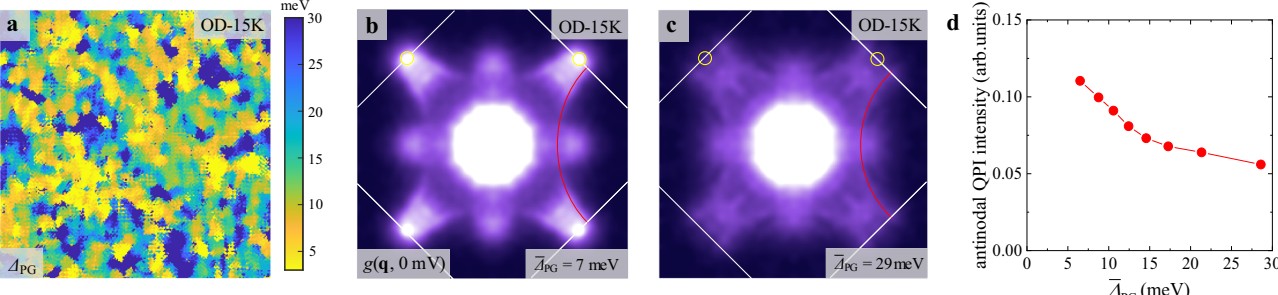

**Fig. 5 | The spatial distribution of normal carrier QPI. a** The spatial distribution of pseudogap in the OD-15K sample shows a typical nano-patch feature. **b and c** The zero-bias QPI patterns in regions with averaged $\bar{\Delta}_{PG} = 7$ meV, and $\bar{\Delta}_{PG} = 29$ meV,

respectively. **d** The evolution of zero-bias QPI intensity near antinodes with the averaged local pseudogap size.

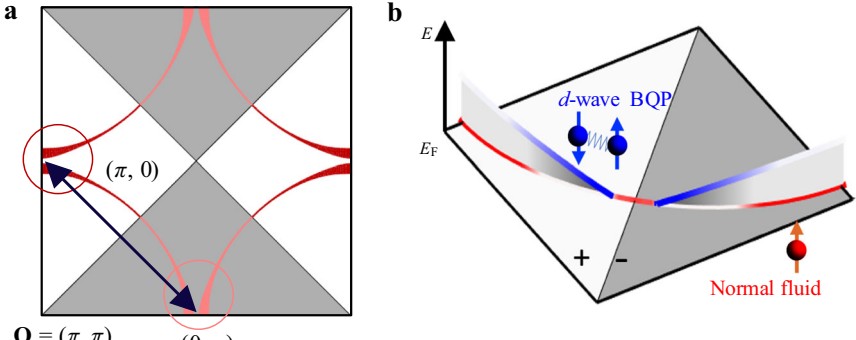

**Fig. 6 | The schematic diagram of low-energy excitation in overdoped cuprate. a** Schematic diagram of the pair-breaking scattering process with sign-changing superconducting order parameter between two antinodes. The vHS enhances such scattering due to the large joint DOS. **b** Schematic diagram of the low-energy

excitations in overdoped cuprates. In addition to that, near the nodal region with vanishing $\Delta_{SC}$, the pair-breaking scattering in the antinodal region creates a branch of normal fluid.

---

trend of different samples with varied doping levels (Fig. 3j). Interestingly, the larger $\Delta_{PG}$ regions show the arc-like QPI patterns with smaller Luttinger volume, confirming the validity of pseudogap size as a local doping level marker. Therefore, the arc-like zero-bias QPI exists in all areas of an overdoped sample, and is not closely tied to the phase-separated non-superconducting regions because of doping inhomogeneity. Besides, we have directly counted the percentage of normal regions in the OD-15K sample and found that it is merely ~10% (Supplementary Material Section IX); thus is not enough to account for the strong arc-like zero-bias QPI. This is consistent with the finding of a 12% gapless region in similar overdoped Bi-2201 with $T_c = 12$ K in another STM work using a different data analysis method[7]. The normal-carrier QPI features start to appear in the slightly overdoped OD-28K sample, where strongly overdoped normal regions are very scarce, also indicating that the normal carrier QPI is not caused by real-space phase separation.

**Disorder-induced Fermi surface in superconductor**
Although the existence of normal fluid with gapless Fermi sheets in the superconducting ground state of overdoped cuprates has not been stressed in previous experimental reports, it is anticipated in a recent theoretical proposal focused on disorder scattering[9]. As shown by the schematic diagram in Fig. 6a, the cross-antinode scattering is a strong pair-breaking process because of the sign-changing nature and the flat band near the antinodal vHS, thus generating a large scattering rate. When the coherence length of disorder scattering $\xi_{scatter}$ is comparable to the superconducting coherence length $\xi_{SC}$, which is equivalent to the condition $\hbar/\tau \sim \Delta_{SC}$, gapless normal carriers appear and generate

substantial unpaired electrons even in the superconducting ground state. To simulate the situations studied in this work, we use the self-consistent mean-field method to evaluate a $d$-wave superconductor with non-magnetic disorder scattering (see Supplementary Material Section X for the details of the simulation). As shown in Fig. 7b for the underdoped case, the zero-energy electronic spectral function in the superconducting state mainly consists of nodal quasiparticles due to the vanishing of the superconducting gap at the node. The full Fermi surface recovers when the system enters the normal state in Fig. 7a. In contrast, Fig. 7c, d reveal distinctly different behavior in the overdoped case with more disorder scattering. Approaching the vHS generates a large pair-breaking scattering rate, resulting in a nearly full Fermi surface even in the superconducting state, which is highly similar to that in the normal state. The strong contrast between the two cases reinforces the deterministic role of disorder scattering in the presence of normal fluid in the superconducting state.

The disorder scattering rate of the quasiparticles extracted from the transport experiment exhibits strong anisotropy, which has the maxima scattering rate at the antinodal region[36]. The antinodal quasiparticle scattering rate in La$_{2-x}$Sr$_x$CuO$_4$, another single-layer compound with similar maximum $T_c$ as Bi-2201, is estimated to be 64 ps$^{-1}$, corresponding to a $\hbar/\tau \sim 42$ meV[36] that meets the condition $\hbar/\tau > \Delta_{SC}$. It indicates that the antinodal quasiparticle in overdoped cuprate lies in the regime that is capable of hosting a gapless Fermi surface, which behaves like a branch of normal carriers. It has also been proposed recently that the apical oxygen vacancy can be responsible for the extensive anisotropic scattering near the antinodal region[11].

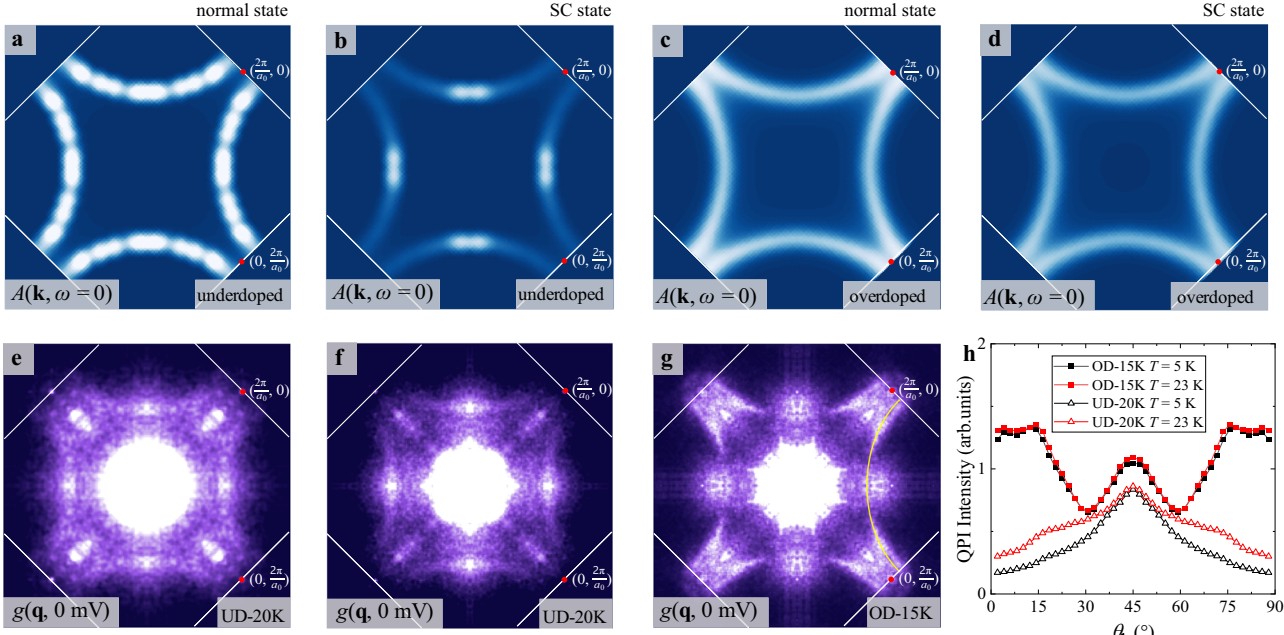

**Fig. 7 | Variable temperature QPI and model calculations based on disordered *d*-wave superconductor. a** and **b** The calculated spectral function $A(\mathbf{k}, \omega)$ at $E_F$ in the normal and superconducting states of underdoped cuprate. The superconducting $A(\mathbf{k}, \omega)$ has residual excitation near the nodes with vanishing $\Delta_{SC}(\mathbf{k})$, while the normal $A(\mathbf{k}, \omega)$ displays a full Fermi surface. **c** and **d** In the overdoped regime, the $A(\mathbf{k}, \omega)$ at $E_F$ in the superconducting state is highly similar to the normal state Fermi surface. **e** and **f** FT of the conductance maps $g(\mathbf{q}, 0\,\text{mV})$ in the UD-20K sample taken at 5 and 23 K, respectively. The QPI pattern in the superconducting state is mainly located near the nodal region, while in the normal state, higher intensity appears near the antinodal region due to thermally excited quasiparticles. **g** FT of the conductance maps $g(\mathbf{q}, 0\,\text{mV})$ in the OD-15K sample at 23 K, which is nearly identical to that at 5 K shown in Fig. 3g. The Brillouin zone boundaries and lattice wavevector are marked by white lines and red points. **h** Angle dependence of the zero-bias QPI intensity below and above $T_c$ for the UD-20K and OD-15K samples, revealing the strong contrast in the underdoped and overdoped regimes.

To test the validity of this picture, we compare the zero-bias QPI on the same field of view of the OD-15K and UD-20K samples at $T = 5$ and 23 K, respectively, taken with the same experimental setup (see Supplementary Material Section XI for details of atomic scale tracking). For the OD-15K sample in Figs. 3g and 7g, the two sets of $g(\mathbf{q}, 0\,\text{mV})$ data are nearly identical, despite the fact that one is deep in the superconducting state while the other is above $T_c$. For the UD-20K sample, in contrast, there is an apparent enhancement of zero-bias QPI from below to above $T_c$ near the antinodal regime, as shown in Fig. 7e, f. The extracted angle dependence of QPI intensity in Fig. 7h reveals the thermally enhanced quasiparticles near the antinodal region in the UD-20K sample and the almost identical superconducting and normal state QPIs in the OD-15K sample. These results are highly consistent with the theoretical simulations (see Supplementary Material Section XII for comparison with theoretical simulations) and the temperature-independent disorder scattering rate extracted from the transport experiment[36], which demonstrate that sizable normal fluid density already exists in the superconducting ground state of overdoped cuprates, as sketched in Fig. 6b.

## Discussion
The nearly temperature-independent features of the emergent normal fluid in overdoped cuprates clearly demonstrate its fundamental difference from the classical two-fluid model[37], which originates from trivial thermal excitations and thus vanishes at low temperatures. Moreover, the quantum normal fluid mainly emerges in the antinodal region with a relatively large superconducting gap, which is also opposite to the expectation for thermally excited normal carriers or the finite instrument resolution because the core physics here is the extensive pair-breaking for the flat antinodal band instead of isotropic thermal excitation. We extracted the superconducting gap

size of the OD-15K sample, which is around 5.9 meV (see Supplementary Material Section XIII for details). The thermal excitation at the experimental temperature $k_B T \sim 0.43$ meV cannot generate a large portion of the normal carrier at the antinodal region with the largest gap.

We have performed a similar experiment on overdoped Bi-2212 with $T_c = 66$ K but find that the arc-like QPI pattern only appears at finite energy (see Supplementary Material Section XIV). It is probably because the large antinodal $\Delta_{SC}$ makes it difficult to meet the condition $\hbar/\tau > \Delta_{SC}$ for generating sufficient normal carriers. This result is also consistent with the observation that low-temperature residual electronic specific heat is absent in overdoped Bi-2212 with $p < 0.22$[38].

The presence of emergent normal fluid in overdoped cuprate provides a natural explanation for the enhanced zero-bias DOS and diminishing Bogoliubov QPI with overdoping, as shown in Figs. 1 and 2, as well as the low temperature residual electronic specific heat, is known as the Somerfield coefficient[22–25]. It can also explain a variety of anomalous experimental observations, including the missing superconducting electrons observed by mutual inductance[4], the two-component fluid transport behavior[39], the uncondensed Drude-like peak at low temperature in overdoped cuprates[5], and the persistent superconducting fluctuation above $T_c$[6], as well as in the overdoped non-superconducting regime[40]. Moreover, it is consistent with the postulation that the Planckian quadrature magneto-resistance in overdoped cuprates is due to the discontinuity of Fermi surface sections induced by the $k$-space distribution of normal carriers[36,39,41,42]. The recent finding of the extension of the superconducting dome to the more overdoped regime in a low-disorder cuprate system is also consistent with this picture[43].

In summary, the zero-bias QPI patterns with arc-like wavevectors demonstrate the existence of normal fluid in the superconducting ground state of overdoped cuprates. The dispersion relation, doping

dependence, energy dependence, and spatial distribution are all consistent with the emergent normal carriers induced by pair-breaking scattering between the antinodal quasiparticles with a sign-changing gap function. Most importantly, our results provide a microscopic mechanism for the suppression of superconductivity in the overdoped regime. The cross antinodal pair-breaking scatterings are responsible for the reduction of superfluid density and suppression of $T_c$, which eventually drive the QSMT[9].

## Methods

### Sample growth

High-quality Bi-2201 single crystals with La or Pb substitutions are grown by the traveling solvent floating zone method and annealed in $O_2$ under different conditions to tune the hole density over a wide range, as described in a previous report[21]. The detailed ingredients are described in Tab. S1.

### STM measurements

The single crystals are cleaved in situ at $T = 77$ K in the ultrahigh vacuum preparation chamber and immediately transferred into the STM stage cooled to $T = 5$ K. The STM topography is collected in the constant current mode with an etched tungsten tip, which is calibrated on a clean Au(111) surface. The differential conductance $dI/dV$ is acquired by a standard lock-in technique. The data supporting the analysis in the main text contain a series of spectral grids on samples with different doping levels, and the detailed parameters are listed in Tab. S2. Before the FT analysis of QPI data, the Lawler–Fujita algorithm[44] is applied on the $dI/dV$ map to guarantee the accuracy of the momentum coordinate. The FT of QPI data are symmetrized along the high-symmetry axis of $CuO_2$ planes, namely the $q_{x,y} = 0$ and $q_x \pm q_y = 0$ axes. The un-symmetrized dataset is displayed in Supplemental Material Section XV, which reveals qualitatively the same patterns as the symmetrized ones.

## Data availability

The source data generated in this study are provided in the Supplementary Information and Source Data file. Source data are provided with this paper.

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

## Acknowledgements

We thank Z.Y. Weng and G.M. Zhang for their helpful discussions. This work was supported by the Basic Science Center Project of NSFC under grant nos. 52388201 and 11888101. Y.W. is supported by the Innovation Program for Quantum Science and Technology (grant No. 2021ZD0302502), and the New Cornerstone Science Foundation through the New Cornerstone Investigator Program and the XPLORER PRIZE. D.H.L. was funded by the US Department of Energy, Office of Science, Office of Basic Energy Sciences, Materials Sciences and Engineering Division, under contract no. DEAC02-05-CH11231 (Quantum Material Program KC2202). X.J.Z. is supported by the Strategic Priority Research Program (B) of the Chinese Academy of Sciences (XDB25000000).

## Author contributions

S.Y. and Y.W. initiated this project. H.Y., C.Y, and Y.C. prepared the single crystal under the supervision of X.Z., S.Y., M.X., C.Z., X.L., and Z.H. carried out the STM experiments under the supervision of Y.W., Z.-X.L., and D.-H.L. performed the numerical simulation. S.Y. and Y.W. prepared the manuscript with comments from all authors.

## Competing interests

The authors declare no competing interests.
