## [Peer Review File · Nature Communications]

Emergent normal fluid in the superconducting ground state of overdoped cupratesREVIEWER COMMENTS

Reviewer #1 (Remarks to the Author):

The manuscript by Ye et al. is a noteworthy exploration of superconductivity in overdoped cuprates through STM measurements. The authors conduct a systematic examination of Quasiparticle Interference (QPI) patterns across varying doping concentrations, revealing insights into pair-breaking effects induced by disorder scattering. While commendable, a few considerations should be addressed to enhance the manuscript's potential for publication in Nature Communications.

1. The assertion that arc-shaped features in QPI are exclusive to overdoped samples raises concerns. Fig. 2 demonstrates similar patterns in optimally and underdoped samples, albeit on a smaller scale. Exploring the evolution of QPI mappings in optimally and underdoped samples, particularly near energies close to the Fermi level of the overdoped sample, would provide a more comprehensive understanding.

2. The claim of enhanced pair-breaking due to increased disorder concentration requires additional support. While Fig. 5 suggests a correlation between visible features and smaller pseudogap regions, it does not rule out the possibility of coexisting normal carriers. A comparative analysis of the temperature dependence of QPI for overdoped, optimally doped, and underdoped samples would strengthen this claim.

3. Considering the surface-sensitive nature of STM measurements, how confident are the authors that their findings accurately reflect the superconducting orders in the bulk state of the system, especially in the high oxygen deficient regime where surface reconstructions can be significant?

In conclusion, this manuscript presents insightful findings but requires additional analysis and clarification to support its key conclusions. Addressing these concerns would significantly strengthen the manuscript, making it suitable for publication in Nature Communications.

Reviewer #2 (Remarks to the Author):

Ye et al present a detailed scanning tunneling spectroscopy (STS) dataset measured on cleaved Bi-2201 as a function of doping. They study the quasiparticle interference (QPI) pattern over a wide range of hole concentration as a function of energy and temperature – temperature dependent STS is a very demanding experiment, even though the temperatures are not very high in this case. The focus of this work is on the suppression of superconductivity in the overdoped regime. Bi-2201 was chosen to investigate the emergent normal fluid in the overdoped regime because it has a much smaller superconducting gap near the antinodes than e.g. Bi-2212.

The main result of the study is the observation of a QPI wavevector that forms an arc-like pattern present all the way to zero bias. These arcs in momentum space appear at the Fermi level only upon entering the overdoped regime (I note they are present at all doping levels at higher energies). The study shows that the intensity of the arcs at zero bias is increasing with increasing hole concentration. The authors contend that the energy dispersion of these arcs is incompatible with the octet model for d-wave superconductivity but is highly consistent with the scattering interference of gapless normal carriers. The scattering responsible for this emerging normal fluid seems mainly located near the antinodes, is pair breaking (connects regions with opposite signs of the d-wave superconducting gap) and independent of temperature. The authors provide evidence that these unpaired electrons coexist with the superconducting condensate.

Based on these observations, Ye et al conclude that the primary cause for the weakening of the superfluid density and the suppression of T_c in overdoped cuprates is the sign-changing pair-breaking scattering between flat antinodal bands. The authors propose that the phase transition into a non-superconducting state is due to the resulting reduced superfluid density, and not to the closing of the superconducting gap.

The manuscript submitted by Ye et al. is addressing an important open question in high

temperature cuprate superconductors. Their experimental insight into the superconductor-to-metal phase transition in highly overdoped Bi-2201 is of current interest. The detailed experiments and proposed analysis are sound and suitable for Nature Communications.

Before fully supporting the publication of this study, I have a few points I would like the authors to clarify.

1. The normalization of the conductance curves presented in Fig.1b is unusual. Why choose to normalize to the mean DOS at +/- 70mV?
2. In Fig.1b, the authors present the spatially averaged dI/dV spectrum for each sample identified by a nominal doping level. However, panels e to h show a very wide variety of line shapes of the tunneling conductance curves, corresponding most likely to a large spread in local doping levels. Therefore, I am wondering to what extent it makes sense to discuss the average over such a large distribution of curves (and doping) – it amounts to discussing the average tunneling conductance between a superconducting and a pseudogap phase.
3. There is still a small gap visible at the Fermi level in the OD-NSC averaged spectrum in Fig.1b. Any idea what this gap is?
4. What evidence do the authors have to support their claim that the zero-bias DOS is increasing more rapidly with reducing pseudogap size in overdoped samples than in underdoped ones. The scatter of the data points in Fig.1c is too large with too few points to support this claim.
5. The authors write that the superconducting gap size is nearly constant from OP-32K to OD-15K samples. Which feature in their spectra do they identify with the superconducting gap? Considering the superconducting dome shown in Fig.1a, their statement implies that the reduced gap is increasing with doping in overdoped Bi-2201. This would be very different from other observations, including in Bi-2212, where $2\Delta/kT_c$ is getting smaller with increased overdoping to approach the value for a d-wave superconductor (about 4.3).
6. The authors argue that the pseudogap size provides a measure of the local doping level. They also find that the zero-bias arcs are measured in all areas of the overdoped samples. The largest pseudogap regions in these samples presumably correspond to underdoped regions – is this not in contradiction with the statement that the QPI arcs at zero-bias only develop in overdoped Bi-2201?
7. For consistency with the other figures in the paper, I suggest turning Fig.3c and 3i by 45 degrees.

Reviewer #3 (Remarks to the Author):

The manuscript "Emergent normal fluid in the superconducting ground state of overdoped cuprates" presents quasiparticle interference data on hole doped Bi2201 BSCCO samples in support of the presence of gapless antinodal quasiparticles for overdoped samples. The authors claim this represents a branch of normal fluid emerging from pair breaking scattering between flat antinodal bands near the van Hove singularity, and is the primary cause for the reduction in superfluid density and decrease in T_c . High quality QPI data is presented in support of these claims, and in general I believe this extensive and thorough work on a class of samples complementary to the Bi2212 family more extensively studied by STM is a meaningful and important contribution to the discussion of superconductivity in the cuprates, meriting publication in Nature Communications. The detailed look at the fermiology of this class of overdoped cuprates seems to allow for a clearer identification of the mechanism of T_c suppression adding an important aspect to the discussion of high- T_c superconductors. I do have some comments regarding interpretation and clarity of the presentation.

In terms of novelty, given the extensive work done by STM and QPI on Bi2212 samples, I think the authors need to focus on the (impressive!) comprehensiveness of the study they have carried out.

Most notably reference 16 (He et al. Science 2014), explores a similar set of features in Bi2212, with similar conclusions, also finding an arc feature associated with scattering from the antinodal regions in an overdoped sample, alongside increased DOS within the gap. While the authors here distinguish their current study by claiming ref 16 attributes these to antinodal Bogoliubov QPI, this is not my read of the prior work, and in fact ref 16 states in reference to these features: "This autocorrelation corresponds to the low-energy antinodal JDOS in the presence of pair breaking and matches well with the data (compare Fig. 2, K and L to G and H)... We conclude that the extinction of octet QPI at the AFBZ boundary in underdoped Bi2201 (Fig. 2I), followed by the appearance of antinodal QPI at higher doping, reveals the FS reconstruction (1-4) shown schematically in Fig. 2J." and in the conclusion "at high doping the PG coexists with superconductivity at the antinode but correlates with suppressed superconducting coherence, suggesting a competitive relationship". To my reading this echoes the findings here. That the authors have found the same relationship in another class, with a more thorough examination of the fermiology and more detailed doping dependence most certainly adds to the story, however I think the authors should be clear in where their contribution differs from the prior work. This does not diminish the importance of the work, and I believe strengthens the conclusions, however changes the focus of the novel aspects to (i) a related but different material set adding some degree of generality, and (ii) more in-depth study of the fermiology with clear attribution to a branch of normal carriers within the SC state (as the prior work leaves open some ambiguity) and relation of the increased pair-breaking scattering to the vHs.

The opening data discussion focuses on the apparent non-zero DOS at E_f , and relates this to the increasing residual specific heat, with both of these pieces of evidence supporting a picture where gapless normal carriers (arising from pair breaking scattering?) overtake the superconducting fraction. Before using this as a solid form of evidence, I would have like to see a discussion of the thermal (and other sources of) broadening present in the data, and how for narrowing gaps this may contribute to the apparent increase in DOS (especially for a d-wave gap). The authors state that although the pseudogap decreases in size, the superconducting gap remains $\sim 10\text{meV}$, however it does seem like in the OD-15K sample the gap may be a bit narrower. Since the T_c is no longer many times the measurement temperature of 5K, it is difficult to untangle effects of the coherence peak broadening from other appearances to the data, and I would recommend fitting a basic Dynes model to extract the gap magnitude for each T_c to more rigorously make this claim, and aid in extracting a robust residual zero-bias DOS. This would also further emphasize the reduced T_c is not due to decrease in pairing strength/smaller gap, but must be due to another mechanism; i.e. that proposed here due to increased scattering driving a larger competing normal fluid branch.

It would be immensely helpful to the reader for the authors to label the directions/axes in the QPI and CECs presented in figures. For example, labelling the corners/high symmetry directions of the BZ in units of π . In particular, it appears that there is a rotation between figure 2d (fermi surface schematic) and 3c, and 6a-d (fermi surface spectral function)? Also, for those of us not experts in cuprates, labelling clearly the nodal and antinodal directions in both the FS panels and at least one QPI panel per figure would be helpful. Note: I did like the clear labelling of panels with energies, types of signals, etc. labelled within the figure.

The authors have included an extensive SI, which contains quite a lot of detail on data analysis as well as additional data. In particular, the full figure S10 showing topography, real-space dI/dV , and resulting QPI (featured in figure 3) is quite impressive, and contains a great deal of rich information in the real-space plots. Perhaps it is too much to include in the paper, but I would have like to see this in the main text, and some discussion around the checkerboard order, and the quite clear increase in disorder on the overdoped side (perhaps all consistent with the perspective of increased scattering resulting in a larger fraction of normal state electrons). At the very least, I think it would be highly beneficial for the authors to more clearly point to the relevant sections of the SI in the main text (for example in describing fig 1 c,d, fig 3j, and for the corresponding real-space maps for fig 3d-h).

The one item that perhaps should be added to the SI is a description of the processing steps used for the QPI. It is clear from the data presented that the QPI patterns are at least symmetrized,

however this should be clearly stated (how many times folded?) and any other processing done explained.

In this letter we provide a point-to-point response to the reviewers' comments.

In the following, the reviewer's original comments are shown by blue italic characters.

The authors' responses are shown by black normal characters.

Reviewer #1 (Remarks to the Author):

The manuscript by Ye et al. is a noteworthy exploration of superconductivity in overdoped cuprates through STM measurements. The authors conduct a systematic examination of Quasiparticle Interference (QPI) patterns across varying doping concentrations, revealing insights into pair-breaking effects induced by disorder scattering. While commendable, a few considerations should be addressed to enhance the manuscript's potential for publication in Nature Communications.

We thank the reviewer for the positive comments about our work. The reviewer's highly thoughtful comments have significantly improved our understanding of the physics and the clarity of the manuscript. In the revised manuscript, we have carefully addressed all the questions and concerns. Below we provide a point-to-point response.

1. The assertion that arc-shaped features in QPI are exclusive to overdoped samples raises concerns. Fig. 2 demonstrates similar patterns in optimally and underdoped samples, albeit on a smaller scale. Exploring the evolution of QPI mappings in optimally and underdoped samples, particularly near energies close to the Fermi level of the overdoped sample, would provide a more comprehensive understanding.

We thank the reviewer for this excellent suggestion. The reviewer is certainly correct that the arc-shaped QPI features are not exclusive to overdoped samples – in fact we did not make that assertion in the manuscript. What we intend to emphasize is that the arc-shaped QPI **at zero bias, corresponding to the Fermi energy (E_F)**, becomes evident only in the overdoped regime. The zero-bias QPI is the most crucial one because it is directly related to the density-of-state at E_F , revealing a branch of normal carrier with gapless Fermi surface. Below in Fig. R1 we present more complete energy-dependent QPI results on the underdoped and optimally doped samples. The arc-shaped patterns are weak but evident at high energies in the UD-20K and OP-32K samples. The QPI at negative biases of the two samples are similar to that at the E_F of overdoped samples, considering the shift of E_F with increasing hole doping.

These comparisons indeed provide a more comprehensive understanding of the QPI phenomena in cuprate. The arc-like QPI patterns at finite biases in the normal state of

underdoped samples indicate that they derive from the normal carrier band structure and are not exclusive to the overdoped one. However, the finding that arc-shaped QPI exists at E_F of the overdoped sample indicate the breaking of Cooper pairs and the recovery of normal carrier band even in the superconducting ground state.

We appreciate the insight of the reviewer, which helps further strengthen our argument. We have added a new Supplementary Session SVI to show the data in Fig. R1 and the discussion about arc-shaped QPI at negative biases of the underdoped and optimally doped samples. We also revised the main text accordingly and clearly emphasize the bias dependence of arc-shaped QPI in cuprates with different hole densities in page 11, line 10.

Fig. R1| QPI patterns of the UD-20K and OP-32K samples. a-f, The FT of conductance maps at -30, -20, -10, 10, 20 and 25 mV in UD-20K sample. **g-l,** The FT of conductance maps at -30, -20, -10, 10, 20 and 25 mV in OP-32K sample.

2. The claim of enhanced pair-breaking due to increased disorder concentration requires additional support. While Fig. 5 suggests a correlation between visible features and smaller pseudogap regions, it does not rule out the possibility of coexisting normal carriers. A comparative analysis of the temperature dependence of QPI for overdoped, optimally doped, and underdoped samples would strengthen this claim.

We thank the reviewer for raising these issues. In Fig. 5d and Fig. S4, our main point is the normal carrier contribution is enhanced with decreasing pseudogap size hence increasing hole density. In the Bi-2201 samples studied here, the doping level is not simply proportional to disorder concentration because there are different types of dopants, including interstitial oxygen, apical oxygen vacancy and Pb substitution of Bi. Moreover, our results show that the normal carriers do coexist with the superconducting condensate in the ground state because the arc-like QPIs are present at zero-bias and over the whole sample. We agree with the reviewer that a comparative analysis of the temperature dependence of QPI will strengthen our main conclusion. In Fig. R2 below, we quantitatively compare the QPI intensity below and above T_c in the underdoped UD-20K and overdoped OD-15K samples. The QPI intensity in OD-15K is independent of temperature, which supports the disorder scattering mechanism. In UD-20K, in contrast, the zero-bias QPI is significantly enhanced in the normal state due to the pair breaking by thermal activation. This comparative result is consistent with our theoretical simulation, as discussed in Supplementary Session SXII.

Fig. R2| a-b, FT of the conductance maps $g(\mathbf{q}, 0 \text{ mV})$ in the OD-15K sample taken at 5 K and 23 K, respectively. **c-d**, FT of the conductance maps $g(\mathbf{q}, 0 \text{ mV})$ in the UD-20K sample taken at 5 K and 23 K, respectively. **e**, Angle dependence of the zero-bias QPI intensity below and above T_c for the UD-20K and OD-15K samples, revealing the strong contrast in the underdoped and overdoped regimes.

Inspired by the reviewer's comments, we have performed a more thorough analysis by using the atomic-precision registry of our datasets to extract and compare the spatial distribution of zero-bias QPI intensity at $T = 5 \text{ K}$ and 23 K in the OD-15K and UD-20K samples. As shown in Fig. R3a, the zero-bias QPI distribution in OD-15K converge in a same trend below and above T_c , but zero-bias QPI intensity in UD-20K are strongly enhanced by heating

up to the normal state. It reinforces our conclusion that the emergent normal fluid in the overdoped regime is generated by disorder scattering, which is temperature independent.

Fig. R3| a-b, The zero-bias QPI intensity near the antinodes at $T = 5$ K and 23 K with respect to the averaged local pseudogap size in the OD-15K and UD-20K samples, respectively.

3. Considering the surface-sensitive nature of STM measurements, how confident are the authors that their findings accurately reflect the superconducting orders in the bulk state of the system, especially in the high oxygen deficient regime where surface reconstructions can be significant?

It is true that STM is surface sensitive and the results may not necessarily reflect the bulk property. Over the past three decades, the community has reached a good understanding of this issue so that a careful choice of material can circumvent this problem. The Bi-2201 compound belongs to the Bi-family cuprates, which are known to have a weak van der Waals bonding between two neighboring Bi-O layers. Therefore, the electronic structure of the cleaved surface remains consistent with that in the bulk because the cleaving does not disrupt any strong bonds (*Rev. Mod. Phys.* 79, 353 (2007)). This is why the Bi-family cuprate has been the prime choice for surface sensitive probes such as STM and ARPES.

Moreover, in recent years Yuanbo Zhang's group has successfully fabricated monolayer Bi-2212 on SiO₂ wafers. Both the STM and transport experiments demonstrate that such a monolayer exhibit the same phenomena as that in the bulk (*Nature* 575, 156 (2019)). This comparison provides compelling evidence that the surface electronic structure of the Bi-family cuprates faithfully reflect the bulk properties.

In conclusion, this manuscript presents insightful findings but requires additional analysis and clarification to support its key conclusions. Addressing these concerns would significantly strengthen the manuscript, making it suitable for publication in Nature Communications.

We thank the referee again for the insightful comments, which help us improve the quality of the manuscript.

Reviewer #2 (Remarks to the Author):

Ye et al present a detailed scanning tunneling spectroscopy (STS) dataset measured on cleaved Bi-2201 as a function of doping. They study the quasiparticle interference (QPI) pattern over a wide range of hole concentration as a function of energy and temperature – temperature dependent STS is a very demanding experiment, even though the temperatures are not very high in this case. The focus of this work is on the suppression of superconductivity in the overdoped regime. Bi-2201 was chosen to investigate the emergent normal fluid in the overdoped regime because it has a much smaller superconducting gap near the antinodes than e.g. Bi-2212.

The main result of the study is the observation of a QPI wavevector that forms an arc-like pattern present all the way to zero bias. These arcs in momentum space appear at the Fermi level only upon entering the overdoped regime (I note they are present at all doping levels at higher energies). The study shows that the intensity of the arcs at zero bias is increasing with increasing hole concentration. The authors contend that the energy dispersion of these arcs is incompatible with the octet model for d-wave superconductivity but is highly consistent with the scattering interference of gapless normal carriers. The scattering responsible for this emerging normal fluid seems mainly located near the antinodes, is pair breaking (connects regions with opposite signs of the d-wave superconducting gap) and independent of temperature. The authors provide evidence that these unpaired electrons coexist with the superconducting condensate.

Based on these observations, Ye et al conclude that the primary cause for the weakening of the superfluid density and the suppression of T_c in overdoped cuprates is the sign-changing pair-breaking scattering between flat antinodal bands. The authors propose that the phase transition into a non-superconducting state is due to the resulting reduced superfluid density, and not to the closing of the superconducting gap.

The manuscript submitted by Ye et al. is addressing an important open question in high temperature cuprate superconductors. Their experimental insight into the superconductor-to-

metal phase transition in highly overdoped Bi-2201 is of current interest. The detailed experiments and proposed analysis are sound and suitable for Nature Communications.

We thank the reviewer for the highly positive comments about our work. The reviewer's thoughtful comments have significantly improved our understanding of the physics and the clarity of the manuscript. In the revised manuscript, we have carefully addressed all the questions and concerns. Below we provide a point-to-point response.

Before fully supporting the publication of this study, I have a few points I would like the authors to clarify.

1. The normalization of the conductance curves presented in Fig.1b is unusual. Why choose to normalize to the mean DOS at +/- 70mV?

We thank the reviewer for raising this issue, which we did not explain in our original manuscript. The data in Fig. 1b encompass a diverse range of samples, from OP-32K to OD-NSC, each displaying distinct spectral features. Our aim is to elucidate the evolution of zero-bias DOS from a deep gap to a peak-like van Hove singularity (vHS). To achieve this, we normalized the spectra by the high-energy background that is not strongly affected by the pertinent low-energy physics. We chose the mean DOS at ± 70 mV as the normalization points because the energy scale is greater than the largest pseudogap in all samples here, and the spectrum exhibits rather smooth features in this energy range. The general trend of the spectral evolution is not sensitive to the choice of normalization parameter as long as the energy is large enough. Figure R4 below displays the averaged spectra normalized at ± 60 mV, ± 70 mV and ± 80 mV, which exhibit qualitatively the same trend.

In the revised manuscript, we add a new sentence in the main text in page 5, line 6 and a new Supplementary Session SII to explain this issue.

Fig. R4| Averaged spectra normalized by the mean DOS at different energies. a-c, The averaged spectra normalized by the mean DOS at ± 60 mV, ± 70 mV and ± 80 mV, respectively. The trends are insensitive to the normalization parameter.

2. In Fig.1b, the authors present the spatially averaged dI/dV spectrum for each sample identified by a nominal doping level. However, panels e to h show a very wide variety of line shapes of the tunneling conductance curves, corresponding most likely to a large spread in local doping levels. Therefore, I am wondering to what extent it makes sense to discuss the average over such a large distribution of curves (and doping) – it amounts to discussing the average tunneling conductance between a superconducting and a pseudogap phase.

We thank the reviewer for this insightful comment. Although in Fig. 1e to h the spectra of each sample have a wide variety of line shapes, their weights are not equal and the spectra on the two limits only occupy a small fraction of the sample. The averaged spectrum is still a good representation of the overall electronic structure of the whole sample. After all, although the microscopic inhomogeneity is an intrinsic property of all cuprates, the electronic phase diagram still has a systematic and generic dependence on the average doping level of a macroscopic sample. The averaged dI/dV spectrum is particularly useful when we compare the STM results with other spatially averaged probes such as ARPES, RIXS and optical spectroscopies. Nevertheless, we agree with the reviewer that particular care should be taken when using the averaged spectra. This is why we show the heat map in Fig. 1d and the wide distribution of spectra in each sample. Combining these figures together, the readers can gain a comprehensive picture regarding the electronic structure in each sample.

In the revised manuscript we add a new sentence to specifically address this issue in page 6, line 12.

3. There is still a small gap visible at the Fermi level in the OD-NSC averaged spectrum in Fig.1b. Any idea what this gap is?

We appreciate the sharp observation of the reviewer of the detailed spectral feature. In fact, we have observed this phenomenon a few years ago in our previous report focusing on the overdoped non-superconducting regime (*New J. Phys.* 20, 063041 (2018)). As shown below in Fig. R5, the spectra of OD-NSC exhibits strong spatial inhomogeneity, containing the vHS peaks and gaps around E_F reminiscent of the pseudogap. We ascribed the gap features to pseudogap for two reasons. First, the gap size is rather large and has strong spatial inhomogeneity. Second, the gap is not particle-hole symmetric with respect to E_F . Both features are inconsistent with the expectation for superconducting gap, such as that shown by Eric Hudson's group in overdoped Bi-2201 (*Nat. Phys.* 3, 802 (2007)). However, we notice that there are recent reports of residual gap features in the overdoped non-superconducting regime of Bi-2201 and $\text{La}_{2-x}\text{Sr}_x\text{CuO}_4$, which are ascribed to the persistent superconducting fluctuation (*Phys. Rev. B* 106, 224515 (2022), *Nat. Mater.* 22, 703 (2023)). The nature of the small gap

thus remains to be completely solved. It is certainly a very interesting issue, but is not the main point of the current manuscript.

Fig. R5| The gap feature in OD-NSC sample. a, Topographic image of an overdoped non-superconducting sample. **b,** The dI/dV spectra taken at different locations indicated in **a**.

4. What evidence do the authors have to support their claim that the zero-bias DOS is increasing more rapidly with reducing pseudogap size in overdoped samples than in underdoped ones. The scatter of the data points in Fig.1c is too large with too few points to support this claim.

Although we did not explicitly make such a claim in our paper, we notice that it is actually an interesting observation implied by our data, especially the heat map in Fig. 1d and the spectral distributions in Figs. 1e-1h. Thanks to the reviewer's question, we gain a deeper understanding of this issue by a more thorough analysis of the data. As shown below in Fig. R6, we establish the relation between the local zero-bias DOS and the pseudogap size of each sample and summarize that into a heatmap. It is apparent that for UD-20K and OP-32K samples, the zero-bias DOS is nearly a constant for different pseudogap sizes. Upon entering the overdoped regime, the zero-bias DOS shows a pronounced increase with decreasing pseudogap size, in addition to the increase of the averaged values. Such a trend is consistent with the physical picture that the pair breaking scattering becomes more significant with overdoping.

In the revised manuscript, we add a new Supplementary Session SIII and a new sentence in page 6, line 22 to discuss this trend.

Fig. R6 | **a-d**, The heatmap of the zero-bias DOS and the pseudogap size for UD-20K, OP-32K, OD-28K and OD-15K samples. **e-h**, The averaged spectral sorted by the pseudogap size for the four samples.

5. The authors write that the superconducting gap size is nearly constant from OP-32K to OD-15K samples. Which feature in their spectra do they identify with the superconducting gap? Considering the superconducting dome shown in Fig.1a, their statement implies that the reduced gap is increasing with doping in overdoped Bi-2201. This would be very different from other observations, including in Bi-2212, where $2\Delta/kT_c$ is getting smaller with increased overdoping to approach the value for a *d*-wave superconductor (about 4.3).

We thank the reviewer for the careful reading of our manuscript. The superconducting gap feature is more evident in the area with larger pseudogap, as indicated by the arrows in Fig. 1a-d. To clearly show the superconducting gap feature in the spectra shown in Fig. 1e-h, we use the second derivative to enhance the gap features (*Nat. Phys.* 14, 1178 (2018); *Nature* 599, 222 (2021)). As shown in Fig. R7e-h, the superconducting gap features are evident from the coherence peak features located 7~9 mV on both sides of the E_F . The gap size shrinks much slower than the decreasing of T_c . It has been reported that the T_c in overdoped $\text{La}_{2-x}\text{Sr}_x\text{CuO}_4$ is close related to the phase stiffness (*Nature* 536, 309 (2016)). The superconducting gap extracted by the *d*-wave Dynes formula fitting in overdoped Bi-2201 is also significantly larger than the BCS prediction and varies slowly contrast to the T_c dropping in overdoped regime (*Nat. Mater.* 22, 703 (2023)). Our results are in general consistent with these results and physical pictures.

We have revised the description of the superconducting gap feature in the clustering spectra in page 6, line 24.

Fig. R7 | **a-d**, The averaged spectra sorted by the pseudogap size in OP-32K, OD-28K, OD-24K, and OD-15K samples. The black arrows indicate the SC gap in the spectra with large pseudogap. **e-f**, The second derivatives of the spectra in **a-d**, showing particle-hole symmetric coherence peaks characteristic of superconducting gaps.

6. The authors argue that the pseudogap size provides a measure of the local doping level. They also find that the zero-bias arcs are measured in all areas of the overdoped samples. The largest pseudogap regions in these samples presumably correspond to underdoped regions – is this not in contradiction with the statement that the QPI arcs at zero-bias only develop in overdoped Bi-2201?

We thank the reviewer for this critical issue. Regarding the inhomogeneities of local doping level, we would like to clarify two key points.

Firstly, we show in Fig. R8 below that the Luttinger volume shrinks gradually with increasing pseudogap size (the arc tips expand to greater distances), which validates the relation between local doping level and pseudogap size. Comparing to the spatially averaged QPI pattern, the Luttinger volume of the area with largest pseudogap (most underdoped) in the OD-15K sample still has a slightly overdoped band structure. Therefore, the presence of normal carrier QPI in all areas of this sample is consistent with the conclusions drawn from spatially averaged experiments.

Secondly, we emphasize that the entire sample is NOT a mere summation of nano-patches with different doping levels. It has been proposed that the proximity effect in cuprate can help

nano-patches with smaller gap survive to higher temperature because they are surrounded by large gap areas (*Phys. Rev. Lett.* 104, 117001 (2010)). Recently, inverse proximity effect has been proposed that the superconducting nano-patch may be influenced by the surrounding non-superconducting metal environment (*Nat. Mater.* 22, 703 (2023)). These arguments underscore the notion that the quasiparticles within a nano-patch are significantly influenced by the global properties of the sample.

Given these arguments, the properties of a nano-patch are NOT solely determined by the local pseudogap size, which can vary at very short length scale. The QPI phenomenon inherently reflects the behavior of quasiparticles over an extended length scale, consistent with the sharp arc-like wavevectors. Therefore, the observation of arc-like QPI in all areas of an overdoped sample is not in contradiction to our physical picture.

We agree that the terminology ‘*proxy*’ might be too strong. We have revised our description accordingly in page 11, line 20 to avoid any misleading statement.

Fig. S8 | Normal carrier QPI in different areas. **a**, The distribution of local pseudogap in the OD-15K sample. **b-i**, The zero-bias QPI pattern in different pseudogap size clusterings.

7. For consistency with the other figures in the paper, I suggest turning Fig.3c and 3i by 45 degrees.

We thank the reviewer for the nice suggestion. We have adjusted Fig. 3c and 3i in the revised manuscript for consistency.

Reviewer #3 (Remarks to the Author):

The manuscript “Emergent normal fluid in the superconducting ground state of overdoped cuprates” presents quasiparticle interference data on hole doped Bi2201 BSCCO samples in support of the presence of gapless antinodal quasiparticles for overdoped samples. The authors claim this represents a branch of normal fluid emerging from pair breaking scattering between flat antinodal bands near the van Hove singularity, and is the primary cause for the reduction in superfluid density and decrease in T_c . High quality QPI data is presented in support of these claims, and in general I believe this extensive and thorough work on a class of samples complementary to the Bi2212 family more extensively studied by STM is a meaningful and important contribution to the discussion of superconductivity in the cuprates, meriting publication in Nature Communications. The detailed look at the fermiology of this class of overdoped cuprates seems to allow for a clearer identification of the mechanism of T_c suppression adding an important aspect to the discussion of high- T_c superconductors. I do have some comments regarding interpretation and clarity of the presentation.

We thank the reviewer for the highly positive comments about our work and the thoughtful comments. In the revised manuscript, we have carefully addressed all the issues. Below we provide a point-to-point response.

In terms of novelty, given the extensive work done by STM and QPI on Bi2212 samples, I think the authors need to focus on the (impressive!) comprehensiveness of the study they have carried out. Most notably reference 16 (He et al. Science 2014), explores a similar set of features in Bi2212, with similar conclusions, also finding an arc feature associated with scattering from the antinodal regions in an overdoped sample, alongside increased DOS within the gap. While the authors here distinguish their current study by claiming ref 16 attributes these to antinodal Bogoliubov QPI, this is not my read of the prior work, and in fact ref 16 states in reference to these features: “This autocorrelation corresponds to the low-energy antinodal JDOS in the presence of pair breaking and matches well with the data (compare Fig. 2, K and L to G and H)... We conclude that the extinction of octet QPI at the AFBZ boundary in underdoped Bi2201 (Fig. 2I), followed by the appearance of antinodal QPI at higher doping, reveals the FS

reconstruction (1–4) shown schematically in Fig. 2J.” and in the conclusion “at high doping the PG coexists with superconductivity at the antinode but correlates with suppressed superconducting coherence, suggesting a competitive relationship”. To my reading this echoes the findings here. That the authors have found the same relationship in another class, with a more thorough examination of the fermiology and more detailed doping dependence most certainly adds to the story, however I think the authors should be clear in where their contribution differs from the prior work. This does not diminish the importance of the work, and I believe strengthens the conclusions, however changes the focus of the novel aspects to (i) a related but different material set adding some degree of generality, and (ii) more in-depth study of the fermiology with clear attribution to a branch of normal carriers within the SC state (as the prior work leaves open some ambiguity) and relation of the increased pair-breaking scattering to the vHs.

We thank the reviewer for these extremely helpful suggestions. With due respect, we still wish to give a short explanation why we summarized the viewpoint of Ref. 16 as “*antinodal Bogoliubov QPP*”. It was mainly from these statements in the paper when referring to the so-called triplet QPI feature: “*It could be considered as a continuation of octet QPI, and we refer to it hereafter as the antinodal QPP*”. The octet QPI is a celebrated model of the QPI between Bogoliubov quasiparticle with banana-shaped equal-energy contours in *d*-wave superconductor (*Phys. Rev. B* 67, 020511 (2003)), so we believe the interpretation is different from ours. But we totally agree with the reviewer that we should put more focus on the novelty of our own results, rather than commenting on previous reports.

There are two important novelties in data analysis that enable us to provide “*more in-depth study of the fermiology*” in cuprates. Our paper is the first to emphasize the importance of the zero-bias QPI in cuprate, which reflects the gapless quasiparticles that are usually absent in the superconducting state. The systematic evolution with doping and temperature of the arc-like zero-bias QPI strongly indicate that the normal carriers coexist with the superconducting condensate in the ground state of overdoped regime. Moreover, we find that the arc-like QPI feature survives to energies higher than the pseudogap scale and disperses like a normal band structure, as shown in Fig. 4. This feature was missing in previous studies, including Ref. 16 which states that “*This feature exists at multiple sub-PG energies without apparent dispersion (fig. S3)*”. The reason is because these previous reports plotted the ratio map $Z(\mathbf{r}, E) = g(\mathbf{r}, E) / g(\mathbf{r}, -E)$ to enhance the superconducting Bogoliubov QPI signals (*Phys. Rev. X* 9, 021021 (2019)). For normal carriers, the band structure has a dispersion without particle-hole symmetry around E_F . Thus our presentation of the raw $g(\mathbf{r}, E)$ map is more meaningful and allows us to reveal the full dispersion of the normal band structure for varied hole densities.

Following the reviewer's suggestions, we have revised the paragraph about the comparison with Ref. 16 and move it to page 11, line 10 after the discussion of Fig. 4.

The opening data discussion focuses on the apparent non-zero DOS at E_f , and relates this to the increasing residual specific heat, with both of these pieces of evidence supporting a picture where gapless normal carriers (arising from pair breaking scattering?) overtake the superconducting fraction. Before using this as a solid form of evidence, I would have like to see a discussion of the thermal (and other sources of) broadening present in the data, and how for narrowing gaps this may contribute to the apparent increase in DOS (especially for a d -wave gap). The authors state that although the pseudogap decreases in size, the superconducting gap remains $\sim 10\text{meV}$, however it does seem like in the OD-15K sample the gap may be a bit narrower. Since the T_c is no longer many times the measurement temperature of 5K, it is difficult to untangle effects of the coherence peak broadening from other appearances to the data, and I would recommend fitting a basic Dynes model to extract the gap magnitude for each T_c to more rigorously make this claim, and aid in extracting a robust residual zero-bias DOS. This would also further emphasize the reduced T_c is not due to decrease in pairing strength/smaller gap, but must be due to another mechanism; i.e. that proposed here due to increased scattering driving a larger competing normal fluid branch.

We thank the reviewer for bringing up the thermal broadening issue. The zero-bias arc-like QPI observed at the antinodal region has a fundamental difference from the normal fluid generated by thermal activation. Firstly, because of the d -wave symmetry, the nodal quasiparticles with smallest superconducting gap are more susceptible to thermal activation. But as shown in Fig. 2, the weight of arc-like QPI is mainly located in the antinodal region with the largest superconducting gap, which is opposite to the expectation for thermally excited normal fluid. Secondly, we have carried out variable-temperature experiment with atomic scale registry, and the zero-bias QPI intensity between $T = 5\text{ K}$ and 23 K in OD-15K sample is nearly the same. This is also in sharp contrast to the expectation of thermal broadening.

To give an accurate estimation of the superconducting gap in the whole sample, we use a d -wave Dynes formula to fit each spectrum in a dense grid. Due to the existence of vHS, the normal DOS background would affect the bare Dynes formula. To reduce the influence of the background, we divide each spectrum at $T = 5\text{ K}$ in the OD-15K sample by the normal state spectrum on the same atomic site. The normalized spectrum is then fitted by the d -wave Dynes formula with thermal and instrumental broadening in the least square method, expressed as:

$$\text{DOS}(E) \propto \int_0^{2\pi} \text{Re} \left\{ \frac{E + i\Gamma}{\sqrt{(E + i\Gamma)^2 - \Delta_{\mathbf{k}}^2}} \right\} d\theta_{\mathbf{k}}$$

Here $\Delta_{\mathbf{k}}$ has the standard d -wave form $\Delta_{\mathbf{k}} = \Delta(\cos k_x - \cos k_y)/2$, where Δ is the gap size at antinodes. The extracted superconducting gap size Δ and quasiparticle scattering rate Γ are displayed in Fig. R9 below. The superconducting gap size centers around 5.9 meV, which is consistent with previous report (*Nat. Phys.* 3, 802 (2007)). The thermal excitation energy of experimental temperature around 0.43 meV cannot be responsible for the presence of emergent normal fluid at the antinodal region. The gap to T_c ratio $2\Delta/k_B T_c$ is around 9.1, much larger than the d -wave BCS value of 4.3. Besides, the quasiparticle broadening evaluates the scattering rate of the quasiparticle $\Gamma \sim \hbar/\tau$, which is centered at 5.5 meV. This scattering rate meets the $\hbar/\tau \sim \Delta_{SC}$ condition for generating gapless quasiparticles.

To address the issues raised by the reviewer, we have revised the main text with corresponding discussions in page 16, line 3 and added a new Supplementary Session SXIII to display the analysis in Fig. R9.

Fig. R9| Dynes formula fitting in the OD-15K sample. a,c, The spatial distribution of the gap size Δ and quasiparticle broadening Γ , respectively. **b,d,** The histogram of Δ and Γ , and their gaussian fits.

It would be immensely helpful to the reader for the authors to label the directions/axes in the QPI and CECs presented in figures. For example, labelling the corners/high symmetry directions of the BZ in units of π . In particular, it appears that there is a rotation between figure 2d (fermi surface schematic) and 3c, and 6a-d (fermi surface spectral function)? Also,

for those of us not experts in cuprates, labelling clearly the nodal and antinodal directions in both the FS panels and at least one QPI panel per figure would be helpful. Note: I did like the clear labelling of panels with energies, types of signals, etc. labelled within the figure.

We thank the reviewer for the very nice suggestions. In the revised manuscript we have adjusted the k -space figures to have the same orientation. We also added more easily understandable Brillouin zone markings. These changes indeed make our article clearer and more accessible to the general readers.

The authors have included an extensive SI, which contains quite a lot of detail on data analysis as well as additional data. In particular, the full figure S10 showing topography, real-space dI/dV , and resulting QPI (featured in figure 3) is quite impressive, and contains a great deal of rich information in the real-space plots. Perhaps it is too much to include in the paper, but I would have like to see this in the main text, and some discussion around the checkerboard order, and the quite clear increase in disorder on the overdoped side (perhaps all consistent with the perspective of increased scattering resulting in a larger fraction of normal state electrons). At the very least, I think it would be highly beneficial for the authors to more clearly point to the relevant sections of the SI in the main text (for example in describing fig 1 c,d, fig 3j, and for the corresponding real-space maps for fig 3d-h).

We thank the reviewer for several excellent suggestions regarding the presentation of the data. We totally agree with the reviewer that the charge order and spatial disorder are important issues that we can address using our data. Following these advices, in the revised manuscript we provide more in-depth discussions in page 6, line 2 about the checkerboard order, which gradually diminishes into short-range glassy patterns in the overdoping process, consistent with previous reports by various experimental techniques. We also give more thorough description and explanation for the data in the supplementary sessions. Following the reviewer's suggestion, we moved the original supplementary Fig. S4 to Fig. 1i-l in the main text to give a direct display of real-space electronic structures.

The one item that perhaps should be added to the SI is a description of the processing steps used for the QPI. It is clear from the data presented that the QPI patterns are at least symmetrized, however this should be clearly stated (how many times folded?) and any other processing done explained.

We thank the reviewer for the thoughtful comment on the description of methodology. In the initial manuscript, we have already described the processing method for QPI data in **Method** of the main text. It includes the correction of the Lawler-Fujita algorithm and symmetrization along lattice high-symmetry directions, which are common and standard

practices in cuprate QPI analysis (*Science* 344, 612 (2014); *Science* 344, 608 (2014); *Nat. Commun.* 10, 1603 (2019)). Following the reviewer's suggestion, we have further refined the description of symmetrization that the folding is conducted along the $q_{x,y}=0$ and $q_x \pm q_y=0$ axes in **Method**.

Below we briefly summarize the main changes to the manuscript:

- (1) Title: no change.
- (2) Abstract: no change.
- (3) Main text: we made several changes as discussed above in the point-to-point response. The changes in the main text are marked in red.
- (4) Figures and captions: we added a row in Fig. 1 to illustrate the real-space electronic structure and the corresponding caption is added. The changes in the captions are marked in red.
- (5) References: no change.
- (6) Supporting materials: we added three supplementary sessions (II, III, VII, VIII, XIII in the revised numbers) in the supporting materials. The new contents are marked in red.

Yayu Wang

Corresponding author, on behalf of all authors.

REVIEWER COMMENTS

Reviewer #1 (Remarks to the Author):

The changes made in the revised manuscript have significantly enhanced its quality, effectively addressing all my previous comments. I am thoroughly impressed with the improvements implemented, which have bolstered the clarity, coherence, and overall impact of the work. I am now in full support for the publication of this work in Nature Communications.

Reviewer #2 (Remarks to the Author):

The authors have answered all questions and comments by all referees and myself. I am still puzzled by the averaging in Fig.1b of such a wide diversity of spectral profiles. But since this does not impact the main conclusions of the paper, I will not insist on correcting or explaining this better - each reader will be able to make their own opinion.

The paper is presenting a very comprehensive and insightful study of the doping dependent electronic properties of Bi-2201, with a convincing story to explain the weakening of the superconducting phase in overdoped compounds. I do recommend publication in Nature Communications.

Reviewer #3 (Remarks to the Author):

The authors have made a number of changes to the manuscript which largely improve readability and clarity in response to all 3 reviewer reports.

The authors have largely addressed my previous comments, and I recommend publication. I would like to add/clarify a few things, each very addressable:

1. I think my comment in reference to He et al. Science 2014 was not taken in the way intended: it was not my intention to suggest removing the comparison, but rather highlighting that the prior data and interpretation are consistent though incomplete. As noted in my previous comments, my interpretation of this previous paper is that they had proposed similar phenomenology, though had less conclusive data to support this. This seems entirely positive as the current manuscript (as explained in the response) has several much more direct pieces of evidence.
2. The discussion of the zero-bias vs. gap filling in the response is very helpful, but missing one small step to be fully convincing: the authors did a fit and histogram of the parameters for the OD-15K sample with an average gap of 5.9meV, but did not then compare this to say the OP-32K. In the manuscript the $\Delta \sim 9\text{meV}$ is reported, but without details of how this was obtained (I would assume from the earlier referee response that this comes from the second derivative, not the same kind of Dynes equation fit?). Being able to quantitatively compare the gap sizes across these extremes would be very helpful in understanding the different mechanisms for the drop in T_c (gap closing vs. loss of phase stiffness) – the comparison as it is seems to support gap closing and loss of phase stiffness, but comparing gap sizes determined by different methods is often misleading. The discussion of the damping scale relative to the gap is helpful, and I think it would be further beneficial for the authors to clarify between fig 1 (where the STS represents a k-integrated picture including both the antinodal and nodal regions which are still susceptible to small scale thermal broadening) and fig 3 where the authors have analyzed the zero-bias weight specifically in the antinodal direction (which as discussed in the response should not be susceptible to thermal effects). My comment had referred to the former, and I had not quite caught that the later discussion was different.
3. I appreciate the addition of the QPI symmeterization information in the methods, though feel they have included the bare minimum in detail. However, on this read-through I also noticed that the authors' data availability statement "All data are available in the main text or the supplementary materials." Collectively I find this insufficient. Either the data and processing code should be made available in a repository, or a much more thorough description and display of the data is necessary (ideally both are done). The authors should include the raw QPI

(unsymmetrized and otherwise unprocessed FFT) in the SI so that readers can confirm that all features are present before these steps are taken. While I agree the methods used are “common and standard practices” that does not mean they are always best practice or always applicable to a given data set. Showing the data in less processed state allows the reader to actively agree that the processing steps have not introduced any artifacts.

Point 3 here must be addressed in some way, either through data sharing and/or details in the SI regarding the processing of the QPI shown. The others I think would help strengthen the manuscript, with hopefully minimal effort.

In this letter we provide a point-to-point response to the reviewers' comments.

In the following, the reviewer's original comments are shown by blue italic characters.

The authors' responses are shown by black normal characters.

Reviewer #1 (Remarks to the Author):

The changes made in the revised manuscript have significantly enhanced its quality, effectively addressing all my previous comments. I am thoroughly impressed with the improvements implemented, which have bolstered the clarity, coherence, and overall impact of the work. I am now in full support for the publication of this work in Nature Communications.

Reviewer #2 (Remarks to the Author):

The authors have answered all questions and comments by all referees and myself. I am still puzzled by the averaging in Fig.1b of such a wide diversity of spectral profiles. But since this does not impact the main conclusions of the paper, I will not insist on correcting or explaining this better - each reader will be able to make their own opinion.

The paper is presenting a very comprehensive and insightful study of the doping dependent electronic properties of Bi-2201, with a convincing story to explain the weakening of the superconducting phase in overdoped compounds. I do recommend publication in Nature Communications.

Reviewer #3 (Remarks to the Author):

The authors have made a number of changes to the manuscript which largely improve readability and clarity in response to all 3 reviewer reports.

The authors have largely addressed my previous comments, and I recommend publication. I would like to add/clarify a few things, each very addressable:

1. I think my comment in reference to He et al. Science 2014 was not taken in the way intended: it was not my intention to suggest removing the comparison, but rather highlighting that the prior data and interpretation are consistent though incomplete. As noted in my previous comments, my interpretation of this previous paper is that they had proposed similar phenomenology, though had less conclusive data to support this. This seems entirely positive

as the current manuscript (as explained in the response) has several much more direct pieces of evidence.

We thank the reviewer for providing further comments regarding the comparison between our results and that in *He et. al. Science 2014*. Following the reviewer's suggestion, we have revised the comparison in page 11, from line 6 to 10.

2. The discussion of the zero-bias vs. gap filling in the response is very helpful, but missing one small step to be fully convincing: the authors did a fit and histogram of the parameters for the OD-15K sample with an average gap of 5.9meV, but did not then compare this to say the OP-32K. In the manuscript the $\Delta \sim 9\text{meV}$ is reported, but without details of how this was obtained (I would assume from the earlier referee response that this comes from the second derivative, not the same kind of Dynes equation fit?). Being able to quantitatively compare the gap sizes across these extremes would be very helpful in understanding the different mechanisms for the drop in T_c (gap closing vs. loss of phase stiffness) – the comparison as it is seems to support gap closing and loss of phase stiffness, but comparing gap sizes determined by different methods is often misleading. The discussion of the damping scale relative to the gap is helpful, and I think it would be further beneficial for the authors to clarify between fig 1 (where the STS represents a k -integrated picture including both the antinodal and nodal regions which are still susceptible to small scale thermal broadening) and fig 3 where the authors have analyzed the zero-bias weight specifically in the antinodal direction (which as discussed in the response should not be susceptible to thermal effects). My comment had referred to the former, and I had not quite caught that the later discussion was different.

We thank the reviewer for this very helpful comment. In fact, we have also extracted the gap size in OP-32K by the Dynes formula. As shown in Fig. R1e-h, the superconducting gap size is centered at 9.5 meV and the quasiparticle broadening is centered at 2.0 meV, which is in sharp contrast to that in OD-15K sample. And it is consistent with the absence of emergent normal fluid in OP-32K because the condition $\Delta \sim \hbar/\tau$ is not satisfied. The extracted value is also consistent with recent report (*Nat. Mater.* 22, 703 (2023)). We add the simulation of the OP-32K data in Supplementary Fig. S13.

Regarding the comparison between the quasiparticle weight in k -integrated STS and the normal fluid weight extracted from the QPI pattern, we believe the later one is more direct for illustrating the accumulation of normal fluid weight. We have revised the main text to discuss the increase of normal fluid weight in the antinodal region in page 9, line 21.

Fig. R1| Distribution of superconducting gap and quasiparticle broadening in the OD-15K and OP-32K samples. a,c, The spatial distribution of the gap size Δ and quasiparticle broadening Γ in the OD-15K sample. **b,d,** The histogram of the Δ and Γ values in OD-15K sample and the respective gaussian fitting. **e-h,** The same sets of data for the OP-32K sample.

3. I appreciate the addition of the QPI symmeterization information in the methods, though feel they have included the bare minimum in detail. However, on this read-through I also noticed that the authors' data availability statement "All data are available in the main text or the supplementary materials." Collectively I find this insufficient. Either the data and processing code should be made available in a repository, or a much more thorough description and display of the data is necessary (ideally both are done). The authors should include the raw QPI (unsymmeterized and otherwise unprocessed FFT) in the SI so that readers can confirm that all features are present before these steps are taken. While I agree the methods used are "common and standard practices" that does not mean they are always best practice or always applicable to a given data set. Showing the data in less processed state allows the reader to actively agree that the processing steps have not introduced any artifacts.

Point 3 here must be addressed in some way, either through data sharing and/or details in the SI regarding the processing of the QPI shown. The others I think would help strengthen the manuscript, with hopefully minimal effort.

We thank the reviewer for the nice suggestion. We have included the unsymmetrized QPI dataset in Supplemental Material Sec. XV, as also shown below in Fig. R2 for both the unsymmetrized and symmetrized QPI dataset. It clearly shows that the arc-like QPI patterns are already present before the symmetrization. We also add more descriptions about this issue in the Method session.

Fig. R2| The un-symmetrized and symmetrized QPI patterns at zero bias. a-g, Un-symmetrized QPI patterns of all samples. The arc-like QPI pattern emerges in the overdoped regime. **h-n,** The symmetrized QPI patterns.

Below we briefly summarize the main changes to the manuscript:

- (1) Title: no change.
- (2) Abstract: no change.
- (3) Main text: we made several changes as discussed above in the point-to-point response. The changes in the main text are marked in red.
- (4) Figures and captions: no change.
- (5) References: no change.
- (6) Supporting materials: we have added a new Supplemental Material Sec. XV. We added the comparison between OD-15K and OP-32K samples in Supplemental Material Sec. XIII. The changes are marked in red.

Yayu Wang

Corresponding author, on behalf of all authors.

REVIEWERS' COMMENTS

Reviewer #3 (Remarks to the Author):

The authors have addressed my few remaining comments, with some clarifying additions to the text, and additional data shared in the SI. I congratulate the authors on this very nice work.